# The influence of lazy information disclosure on stock price crash risk: Empirical evidence from China

Xiaofei Shi[1,2]* , Xuefen Cao[1], Wenxin Xu[1], Yangshi Hou[1], Liwei Shang[3]

1 School of Business Administration, Hebei University of Economics and Business, Shijiazhuang, China,
2 Research Center for Corporate Governance and Enterprise Growth, Hebei University of Economics and Business, Shijiazhuang, China, 3 International Exchange Center, Hebei University of Economics and Business, Shijiazhuang, China

☯ These authors contributed equally to this work.
* shixiaofei01@hotmail.com

**Data Availability Statement:** All relevant data are within the paper and its Supporting Information files.

**Funding:** This research was funded by National Natural Science Foundation of China [72172063];

## Abstract

Information disclosure is an important way for investors to obtain information, the annual report text carries a lot of information, lazy information disclosure is an important form of information disclosure of the annual report text. This paper takes China's A-share listed companies from 2011 to 2022 as the research sample, takes the annual report text information disclosure form as the entry point, uses the computer text analysis technology to measure the text similarity of the annual report to measure the lazy information disclosure, and explores its impact on stock price crash risk. The results show that there is a positive correlation between the similarity of annual report text and the risk of stock price crash, that is, when the information of annual report text is presented in the form of lazy information disclosure, the risk of stock price crash increases. For companies audited by key auditing institutions, the positive correlation between the similarity of their annual reports and the risk of stock price crash is not significant, indicating that key auditing institutions will weaken the positive correlation between lazy information disclosure and the risk of stock price crash. Further, through external attention and analysis of the time delay of annual report disclosure, it is concluded that the management lacks the opportunity and time to hide the bad news, so it is clear that the lazy information disclosure comes from the business situation "the fact is so". The research conclusion of this paper provides evidence support for the influence of lazy information disclosure on stock price crash risk, and also provides useful reference for regulators to improve information disclosure policies and effectively prevents and resolves stock price crash risk.

## 1 Introduction

A strong and stable capital market plays a key role in ensuring the steady operation of the modern economic system. In the context of the 14th Five-Year Plan, it is clearly pointed out that the basic system of the capital market should be improved, the operating system of the

S&T Program of Hebei [22557607D]; Humanities and Social Science Research Project of Hebei Education Department [SD2022054]; The Project of Corporate Governance and Enterprise Research Center [GS2023G]. The funders had no role in study design, data collection and analysis, decision to publish, or preparation of the manuscript.

**Competing interests:** The authors have declared that no competing interests exist.

capital market should be perfected, and the registration system of stock issuance should be fully implemented to ensure that stock prices accurately reflect the performance of enterprises. Avoiding the occurrence of stock price collapse is the guarantee of effective operation of capital market. At the same time, reasonable and accurate stock prices can optimize the allocation of market resources and promote the healthy development of the capital market. Recently, under the influence of domestic and international political and economic environment, A-share fluctuated, resulting in low sentiment of stock investors, which seriously affected the rights and interests of investors and the normal operation of the capital market. Therefore, to discuss the factors affecting the stock price crash risk of listed companies has become the focus of the theoretical and practical circles. The risk of stock price crash refers to the probability of stock price falling sharply. The current research on the causes of stock price crash risk mainly involves the following two factors. First, based on the theory of information asymmetry, if investors are at an information disadvantage, it is difficult for them to have an in-depth understanding of the company, and easy to make unreasonable judgments on the company's financial condition and operating results, thus overestimating the value of the company. When missing information is disclosed to the market, investors will sell shares en masse, increasing the risk of a crash. Second, based on the agency theory and the "bad news" hiding hypothesis, shareholders pursue the maximization of the company's interests, while the management pursues the maximization of their own interests, and their interests are inconsistent. On this basis, the management will seek its own interests by hiding the bad news. When the bad news reaches a critical point and breaks out, a large number of stocks will be sold by investors and the stock price will fall rapidly, thus increasing the risk of stock price collapse.

By regularly disclosing annual reports, listed companies provide stakeholders with information on their financial status, operating results and future development trends, effectively easing the information asymmetry between companies and stakeholders. International Financial Reporting Standards improve the transparency and comparability of accounting information and effectively improve the quality of accounting information by specifying the disclosure content, rules, forms, etc. [1]. Combining the current development status and practical background, China's accounting standards are committed to formulating high-standard information disclosure policies, effectively improving information asymmetry, preventing fraudulent trading and insider trading, and protecting the rights and interests of stakeholders. At present, the vast majority of listed companies are able to timely release regular and irregular reports, announce major matters at any time, and improve the comprehensiveness and accuracy of information disclosure. However, the information disclosure system needs to be improved in the aspects of the positioning, subject, requirement and quality of information supply. The effective information supply for stakeholders is insufficient, and the identification of civil liability of market false information discharger by relevant laws and regulations is not clear enough. In 2018, Zhongtian Energy was punished for not referencing 2017 financial data when publishing restructuring termination information and continuing to use 2016 financial data. Qian and Zhu (2020) believe that the non-management discussion and analysis part of the annual report of listed companies has the characteristics of template and inertia that does not change with time [2]. If the management uses a large number of annual report information in previous years when disclosing the text information of the annual report, that is, the disclosure information has the characteristics of template and does not change with time inertia, it is called lazy information disclosure. Lazy information disclosure as an important form of information disclosure of the annual report text may have an important impact on the information quality of the annual report text, and the quality of the annual report text information has a crucial decisive role in the judgment of the capital market or external stakeholders. Therefore, it is of great theoretical and practical significance for this study to measure lazy information

disclosure by the similarity of annual report texts and explore its relationship with stock price crash risk.

Based on the above analysis, this paper selects China's A-share listed companies from 2011 to 2022 as the research sample, takes the form of annual report text information disclosure as the entry point, measures the lazy information disclosure through the similarity of annual report text measured by computer text analysis technology, and explores its impact on stock price crash risk. The results show that the similarity of annual report text is positively correlated with the risk of stock price crash, that is, the risk of stock price crash increases when the information of annual report text is presented in the form of lazy information disclosure. For companies audited by key audit institutions, the positive correlation between the similarity of their annual reports and the risk of stock price collapse is not significant, indicating that key audit institutions will weaken the positive correlation between lazy information disclosure and the risk of stock price crash. Through external attention to measure the quality of enterprise information disclosure, found that the management lacks the opportunity to hide the bad news, through the annual report disclosure delay found that the management lacks the time to hide the bad news, according to clear lazy information disclosure from the business situation "the fact is so".

Compared with previous literatures, the theoretical contribution of this paper may be reflected in the following three levels. First, it enriches the research perspective of the annual report text information. This paper extends the research on the text characteristics of listed companies' annual reports from the current information readability, intonation, complexity and management characteristics to the text similarity of annual reports. Second: it enrichs the stock price crash risk factors research. Different from the previous discussion of stock price crash risk mainly through the influence of management, shareholders, investors and external regulators on information asymmetry, this paper explores the influence of lazy information disclosure and stock price crash risk by taking the form of annual report text disclosure as the entry point, and finds that lazy information disclosure will increase the stock price crash risk. Third: it deepens the research on the presentation form of information disclosure in the annual report text. Existing literature on lazy information disclosure is relatively broad. This article provides an in-depth analysis of the sources of inert information disclosure, and clarifies that inert information disclosure originates from business situation "the fact is so" through analyst attention, information disclosure lag, and other means.

## 2 Literature review

### 2.1 Research on influencing factors of stock price crash risk

Existing literature divides the influencing factors of stock price crash risk into internal factors and external factors. Internal factors such as the behavior of managers [3, 4], characteristics of managers [5], quality of accounting information [6], corporate governance [7, 8], mixed ownership reform [9], digital transformation [10] green transformation of enterprises [11], ESG performance [12, 13], etc. External factors such as inflation, unemployment rate, GDP, and exchange rate [14], institutional environment [15], and capital market openness [16], among other macro factors; Macro policies such as industrial policy tools [17], merger of national and local taxes [18], and green credit [19]; Stakeholders such as securities analysts [20] and institutional investors [21, 22]. All the above factors can be explained by the information asymmetry theory [23] and the "bad news" hiding hypothesis [24].

### 2.2 Research the influencing factors of the text features of annual reports

The company's operating condition has an important impact on the information characteristics of the annual report text. When the company is in a good state of development, the text

tone of its annual report is relatively optimistic [25]; When the company's business is more complex, the readability of its annual report text decreases [26]; When there are significant changes in the company's business operations, the management's discussion and analysis have a significant degree of change [27]; When corporate performance declines, disclosure of R&D information will increase [28]. In addition, as the direct person responsible for the disclosure of information in the annual report, the characteristics of the management and their motivations on the characteristics of the annual report text can not be underestimated. When the manager has been in office for a long time, the disclosure of annual report text information content is higher; Diligent managers tend to produce prospectuses with high information content. When the management is motivated by "information ambiguity", he increase the possibility of information manipulation in order to maintain the level of earnings, thus reducing the readability of the annual report. Management strategically manipulates the tone of the annual report to conceal bad news [29]; Management misleads investors by disclosing forward-looking information unrelated to earnings information in order to alleviate the uncertainty of the company's operations. When management is motivated by "information supply", he tends to disclose more textual information to explain and supplement financial information. When the company's information environment is poor, management will disclose more forward-looking information to help investors accurately judge the company's value. When the company faces litigation risks, the management will increase the content of textual information disclosure and enhance investors' confidence in the company's development.

## 2.3 Research the economic consequences of the text features of the annual report

Different characteristics of annual reports will have different impacts on the capital market. Firstly, the text information of annual report with poor readability will increase the information asymmetry between investors and companies [30], reduce the market reaction degree and stock volume [31], and reduce the information efficiency of stock price [32]. High stock price synchronization will result in lower stock liquidity and stock returns in the next period [33]. Wang et al. (2022) found that the worse the readability of the company's annual report, the higher the possibility of stock price crash in the future [34]. When the text information of the annual report is highly readable, the information transmission efficiency will be improved, and the earnings information in the annual report will be better reflected in the stock price, thus inhibiting the management of earnings management. Secondly, the positive or negative tone of the annual report text will convey the information of the company's future performance, cause different reactions in the market [35], cause changes in the stock trading volume [36], and change the volatility of the stock price [24]. The tone of the annual report also affects a company's cost of obtaining capital. A positive tone lowers the cost of equity capital, raises the proportion of debt financing and lowers its cost. However, the manipulation of tone will reduce the transparency of information, and when the hidden bad news is released, the risk of stock price collapse will occur in the whole stock market [37].

In addition, the higher the similarity between management analysis and discussion in different years, the more negative the performance of the capital market [27]. Hao and Su (2014) studied the IPO prospectus and found that the lower the similarity of the IPO prospectus compared with the previous issue, that is, the higher the proportion of effective information reflecting the individual characteristics of the company, the more helpful it is to reflect the real value of the stock and reduce the IPO underpricing level [38]. Tetlock (2011) found that the more similar the current annual report is to the previous one, investors tend to sell stocks, and the overreaction of the capital market to redundant information will lead to the reversal of future

returns [39]. Zhao et al. (2019) found that the boilerplate economic consequences discussed and analyzed by management are discretionary, and boilerplate often leads to negative consequences when the financial risk is high [40]. You et al. (2021) found that the higher the similarity between management discussion and analysis and that of the previous year is, it indicates that the company's management deliberately conceals information, which increases the probability of the listed company being punished for violations. However, non-management discussion and analysis conveys more template information. The lower the similarity, the more unstable the company's operation and the higher the probability of violation [41]. Meng et al. (2017) found that the higher the similarity of the discussion and analysis of the management level of listed companies compared with the previous year, the lower the information content transmitted, which increased the risk of stock price crash to some extent [42]. Wang et al. (2021) further analyzed the outlook and review part of management discussion and analysis and found that the information conveyed in the outlook part could effectively reduce the risk of stock price crash [43]. Song et al. (2022) conducted an in-depth study on the relationship between incremental information disclosure of listed companies and capital market pricing efficiency, and found that the less incremental information disclosure, the higher the synchronicity of stock prices, and the greater the risk of stock price collapse [44].

## 3 Theoretical analysis and research hypothesis

### 3.1 Lazy information disclosure and stock price crash risk

The annual reports of listed companies are the main source of information for a large number of investors to make decisions, and accurate and complete information plays a crucial role in the decision-making process of investors. Given that the annual report carries information about the company's operating status and future development direction, detailed textual explanations help investors accurately assess the company's value, thereby reducing the risk of decision-making errors caused by information asymmetry [45, 46]. Therefore, the information conveyed by the annual report text plays an important role in the stability of stock prices.

Annual report as a specific performance of information disclosure by the company's management, the form of its text information disclosure may have an impact on the characteristics of the text information of the annual report. Because the lazy information disclosure has the characteristics of template and does not change with time inertia, when the text information of the current report is expressed in the form of lazy information disclosure, the text of the company's annual report has higher similarity, that is, the current annual report updates less than the previous year's annual report, and the text is closer to each other. Inert information disclosure indicates that the management largely uses previous annual report information during information disclosure and does not truthfully convey information about the current operating status and future development direction. This means that the current annual report has less updated content compared to previous annual reports, which to some extent reduces the transparency of annual report information. The decline in information transparency provides an opportunity for management to conceal bad news [47]. Although the management needs to regularly disclose the company's financial condition, operating results, and other company specific information in accordance with the "Management Measures for Information Disclosure of Listed Companies". However, it is worth noting that existing research suggests that management, driven by self-interest motivation, may conceal bad news in order to obtain equity incentives [48] and excess returns [3]. At the same time, as the direct responsible person for the annual report text, the management has a certain right to choose the content and timing of information disclosure, especially when the listed company is not doing well or the

investment project risk is high, the management may use their private rights to conceal bad news in order to protect their own interests.

When management adopts lazy information disclosure, the decrease in information transparency and self-interest motivation of management increase the possibility of management hiding bad news. Investors are prone to make unreasonable judgments about the company's value without being familiar with its financial situation, operating results, and development prospects. However, the concealment of bad news requires a certain cost, and when the management cannot bear the cost of concealing bad news, negative news tends to erupt. At this point, investors may sell a large amount of stocks, leading to a sharp drop in stock prices [49]. Based on the above analysis, the following assumptions are proposed:

H1: Assuming other conditions remain unchanged, there is a positive correlation between the similarity of the annual report text and the stock price crash risk, that is, when the text information of the annual report is presented in the form of lazy information disclosure, the stock price crash risk increases.

## 3.2 The regulating role of audit institutions

When the management adopts an inert information disclosure strategy, the similarity of the annual report text is higher, that is, the effective information content carried by the annual report text is lower [42], the integrity of the annual report text information is weakened, and the quality of the annual report text information decreases. As a third party independent of shareholders and management, audit institutions can obtain sufficient and appropriate audit evidence and express audit opinions on the annual reports of listed companies by implementing necessary audit procedures, ensuring that the company's financial statements comply with generally accepted accounting principles [50], providing guarantees for the reliability of annual report information [51, 52]. The scale of key audit institutions is relatively large, and auditors have a high level of professional competence and strong independence. They are able to objectively and fairly evaluate the company's annual reports. On the one hand, the audit opinions issued by key audit institutions have a stronger verification effect, which largely guarantees the reliability of annual report information disclosure [53]. On the other hand, the selection of key audit institutions means that the company is willing to promise to provide reliable annual report information. When the current report is presented in the form of lazy information disclosure, key audit institutions audit can alleviate the information asymmetry between stakeholders and companies by sending positive signals to the outside world. On the one hand, the annual report as a direct channel for investors to understand the listed company, the annual report authenticated by key audit institutions helps investors to make accurate judgments, thus reducing the risk of stock price collapse. On the other hand, the annual report is an important source of analysts' forecasts. The annual report authenticated by key audit institutions helps attract more analysts' attention, improving the accuracy of analysts' forecasts, and then reducing the synchronicity of stock prices and reduce the risk of stock price crash [54]. Based on the above analysis, this paper believes that key audit institutions can reduce the impact of similarity of annual reports on stock price crash risk. Accordingly, this paper proposes the following hypothesis:

H2: Assuming that other conditions remain unchanged, the positive correlation between the similarity of company annual reports audited by key audit institutions and the risk of stock price crash is significantly reduced, that is, key audit institutions will weaken the positive correlation between lazy information disclosure and the risk of stock price crash.

## 4 Research design

### 4.1 Sample selection and data sources

In this paper, China's A-share listed companies from 2011 to 2022 are selected as the initial research samples. The annual reports used in this paper are from Shanghai Stock Exchange, Shenzhen Stock Exchange and Jutide Information network. Among them, the annual reports that cannot be obtained directly are collected and sorted manually; the text similarity data of the annual reports are obtained from the annual reports through Python; other data related to the company are from the database of Guotai 'an. In order to reduce the interference of other factors on the regression results, the initial sample data were processed as follows by referring to existing practices: (1) Annual observation samples of financial companies were excluded; (2) Annual observation samples of companies excluding ST or *ST; (3) A total of 12168 samples were obtained by eliminating the samples with data missing and outliers.

### 4.2 Variable selection and measurement

**4.2.1 Explained variable.**   The risk of a collapse in stock prices. Referring to the practices of Hutton (2009) [23] and other scholars, this paper uses NCSKEW and DUVOL to measure the risk of stock price crash. The higher the value of this coefficient, the greater the risk of a company's stock price collapse. Firstly, the specific rate of return of stocks in week t is calculated by the following regression.

$$R_{i,t} = \alpha_0 + \beta_1 R_{m,t-2} + \beta_2 R_{m,t-1} + \beta_3 R_{m,t} + \beta_4 R_{m,t+1} + \beta_5 R_{m,t+2} + \varepsilon_{i,t} \tag{1}$$

weighted by the circulating market value in week t, and $\varepsilon_{i,j}$ is the part of the return rate of the stock that cannot be explained by the market return rate. The weekly specific rate of return is obtained by the formula $W_{i,t} = \ln(1 + \varepsilon_{i,t})$. Secondly, the negative return skewness coefficient is constructed by the following formula:

$$NCSKEW_{i,t} = -\left[ n(n-1)^{\frac{3}{2}} \sum W_{i,t}^3 \right] / \left[ (n-1)(n-2)(\sum W_{i,t}^2)^{\frac{3}{2}} \right] \tag{2}$$

At the same time, the up and down volatility of stock price returns is constructed to measure the stock price crash risk, $n_u(n_d)$ is the number of weeks in which the weekly specific return of stock i is greater than (less than) the annual average return.

$$DUVOL_{i,t} = \log\{[(n-1)\sum\nolimits_{DOWN} W_{i,t}^2]/[(n-1)\sum\nolimits_{UP} W_{i,t}^2]\} \tag{3}$$

**4.2.2 Explain variables.**   Inert information disclosure: Considering that when the annual report text is presented in the form of inert information disclosure, the similarity of the annual report text is relatively high, so this article uses the similarity of the annual report text to measure the degree of inert information disclosure.This article refers to the research of Zheng and Liu (2022) [55] and adopts a text vectorization method to calculate the similarity of annual report texts. The calculation process is as follows: (1) Common stop words such as Baidu Stop word, Harbin Institute of Technology Stop word, and Sichuan University Machine Intelligence Laboratory words are used to stutter the text; (2) Clean the segmentation results; (3) Calculate the TF-IDF value of words in the text. TF refers to the frequency of words appearing in the text, and IDF refers to the probability of a certain word appearing in all texts. The adjusted word frequency is the product of TF and IDF, giving higher weights to words that better reflect text features; (4) Calculate text similarity using cosine function.

**4.2.3 Regulating variable.** Considering the professional competence and independence of the Big Four international firms are stronger than other firms, this paper defines the "Big Four international firms" as key audit institutions. If the audit institution of the listed company in the current year is an "international Big Four" firm, Big4 is denoted as 1; otherwise, it is denoted as 0.

**4.2.4 Control variables.** Referring to the existing literature, this paper controls the following variables that may affect the risk of stock price crash. (1) Company level: company size (size), property right nature (pro); (2) Financial level: financial leverage (lev), return on total assets (roa), return on equity (roe); (3) Corporate governance level: Board size (boardsize), independence (ind), management shareholding ratio (mngmhldn) and sharesbalance (dual). In addition, the year is controlled in this paper. Table 1 shows the definition of the main variables.

## 4.3 Model design

Based on the high similarity of annual reports when they are presented in the form of lazy information disclosure, this paper adopts the similarity of annual reports to measure lazy information disclosure. Considering that the current annual report is often disclosed before May of the next issue, it takes some time for the information conveyed by the annual report text to be absorbed by the capital market, so research related to the information in the annual report text often lags behind one period of processing [2]. It is worth noting that this study examines the impact of information conveyed in annual report texts on the risk of stock price crashes. The fundamental reason for the risk of stock price crashes is the outbreak of negative news, and the accumulation of negative news also takes some time to be reflected in the market [56]. Therefore, when studying the impact of lazy information disclosure on the risk of stock

**Table 1. Variables definition.**

| Variable type | Variable abbreviation | Variable Definition |
|---|---|---|
| Explained Variables | NCSKEW | Negative return skewness coefficient, refer to Formula (2) |
| | DUVOL | The fluctuation rate of stock price returns, refer to Formula (3) |
| explanatory variable | sim_1 | Similarity between the t-th annual report and the previous annual report |
| Adjusting variable | Big4 | If the auditing agency is the International Big Four, it will be recorded as 1, otherwise it will be recorded as 0 |
| control variables | size | Natural logarithm of total assets at the end of the period |
| | pro | If the actual controller of the company is state-owned, it is recorded as 1, and vice versa, it is recorded as 0 |
| | lev | Ratio of total liabilities to total assets |
| | roa | Ratio of net profit to total assets |
| | roe | Ratio of net profit to net assets |
| | boardsize | Number of directors |
| | ind | The proportion of independent directors to the total number of directors |
| | mngmhldn | The proportion of management shareholding in total issued shares |
| | sharesbalance | Ratio of shareholding ratio between the 2nd to 5th largest shareholder and the 1st largest shareholder |
| | dual | If there is a combination of two positions, it is recorded as 1, otherwise it is recorded as 0 |
| | year | Virtual variable generated based on year |
| | industry | Virtual variables generated by industry |

price crash, variables related to the annual report will be processed two periods behind, in order to avoid the sample size reduction caused by the lag. In the process of data processing, text information variables of annual reports from 2009 to 2020 were matched with dependent variables and control variables from 2011 to 2022. In the regression model, if the similarity coefficient of the annual report text is significantly positive, it indicates that the similarity of the annual report text is positively correlated with the stock price crash risk, that is, when the information of the annual report text is presented in the form of lazy information disclosure, the stock price crash risk increases. Hypothesis H1 is established. If the similarity coefficient of the annual report text is significantly negative, it indicates that hypothesis H1 is not valid.

$$NCSKEW_{i,t} = \alpha_0 + \alpha_1 \times Sim_{i,t-2} + \alpha_2 \times Controls_{i,t} + \varepsilon_{i,t} \quad (4)$$

In order to verify the influence of audit institution quality on lazy information disclosure and stock price crash risk, the samples are divided into two groups: companies audited by Big Four international accounting firms and companies audited by non-big Four international accounting firms, and regression is carried out according to model (4).

## 5 Empirical test

### 5.1 Descriptive statistics

The descriptive statistical results of the variables are shown in Table 2. In order to avoid the impact of outlier on the regression results, this paper conducts tail shrinking at the level of 1% and 99% for each continuous variable. The mean skewness coefficient of negative returns is -0.3388, with a standard deviation of 0.7180. The mean volatility of stock price returns is -0.2227, with a standard deviation of 0.4723, indicating that the overall risk of stock price collapse is relatively low for enterprises, but the risk of stock price collapse is relatively high for individual enterprises. The average similarity of annual report texts is 0.6020, with a standard deviation of 0.1594. The difference between the minimum and maximum values of information similarity in annual report texts is significant, indicating a significant difference in similarity between sample companies in annual report texts, providing data support for empirical research. The descriptive results of the selected control variables are basically consistent with existing research.

**Table 2. Descriptive statistics.**

|  | (1) | (2) | (3) | (4) | (5) | (6) | (7) | (8) |
|---|---|---|---|---|---|---|---|---|
|  | count | mean | sd | min | p10 | p50 | p90 | max |
| sim_1 | 12168 | 0.6020 | 0.1594 | 0.1498 | 0.3916 | 0.6135 | 0.7951 | 0.9379 |
| ncskew | 12168 | -0.3388 | 0.7180 | -2.5465 | -1.2459 | -0.2825 | 0.4718 | 1.5662 |
| duvol | 12168 | -0.2227 | 0.4723 | -1.3921 | -0.8384 | -0.2164 | 0.3765 | 0.9524 |
| size | 12168 | 22.810 | 1.3444 | 20.072 | 21.218 | 22.672 | 24.584 | 26.628 |
| lev | 12168 | 0.4907 | 0.1943 | 0.0772 | 0.2205 | 0.4953 | 0.7461 | 0.9010 |
| roa | 12168 | 0.03251 | 0.05203 | -0.1709 | -0.0108 | 0.02923 | 0.0906 | 0.1873 |
| roe | 12168 | 0.05403 | 0.1285 | -0.6901 | -0.0207 | 0.06346 | 0.1662 | 0.3228 |
| pro | 12168 | 0.5980 | 0.4903 | 0.0000 | 0.0000 | 1.0000 | 1.0000 | 1.0000 |
| boardsize | 12168 | 8.8804 | 1.8239 | 0.0000 | 7.0000 | 9.0000 | 11.0000 | 18.0000 |
| ind | 12168 | 37.411 | 5.6798 | 0.0000 | 33.3300 | 36.360 | 42.860 | 80.0000 |
| mngmhldn | 12168 | 3.1425 | 9.4908 | 0.0000 | 0.0000 | 0.003800 | 8.3092 | 50.612 |
| sharesbalance | 12168 | 0.5846 | 0.5415 | 0.02030 | 0.07270 | 0.4041 | 1.3695 | 2.5048 |
| dual | 12168 | 0.1678 | 0.3737 | 0.0000 | 0.0000 | 0.0000 | 1.0000 | 1.0000 |

## 5.2 Correlation analysis

Table 3 shows the correlation analysis results of the main variables. From Table 3, it can be seen that the two measures of stock price collapse risk, NCSKEW and DUVOL, have a high correlation, indicating consistency between the two. Without considering other factors, the similarity of annual report text (sim_1) is positively correlated with the risk of stock price collapse. In order to further clarify the relationship between the similarity of annual report texts and the risk of stock price collapse, this article incorporates other factors that may affect the risk of stock price collapse and conducts multiple regression.

## 5.3 Regression result analysis

**5.3.1 Inert information disclosure and the risk of stock price collapse.** In the regression results of Table 4, the dependent variables in columns (1) and (2) are the negative return skewness coefficient (NCSKEW), while the dependent variables in columns (3) and (4) are the volatility of stock price up and down returns (DUVOL). Columns (1) and (3) show the results for years and industries not controlled, while columns (2) and (4) show the results for years and industries controlled. From each column of the table, it can be seen that there is a significant positive correlation between the similarity of annual report text and the risk of stock price collapse. That is, when the information in the annual report text is presented in the form of inert information disclosure, the risk of stock price collapse increases, and hypothesis H1 is verified. Specifically, under the control of the year and industry, when the similarity (sim_1) of the annual report text increases by 1 unit compared to the previous period, the negative return skewness coefficient and the fluctuation ratio of upper and lower returns increase by an

**Table 3. Correlation analysis.**

| | ncskew | duvol | sim 1 | size | lev | roa | roe |
|---|---|---|---|---|---|---|---|
| ncskew | 1 | | | | | | |
| duvol | 0.875*** | 1 | | | | | |
| sim 1 | 0.040*** | 0.041*** | 1 | | | | |
| size | -0.036*** | -0.057*** | 0.0110 | 1 | | | |
| lev | -0.038*** | -0.039*** | -0.028*** | 0.428*** | 1 | | |
| roa | 0.037*** | 0.020** | -0.00400 | 0.076*** | -0.355*** | 1 | |
| roe | 0.0130 | -0.00500 | -0.0120 | 0.147*** | -0.190*** | 0.846*** | 1 |
| pro | -0.046*** | -0.039*** | -0.028*** | 0.174*** | 0.143*** | -0.060*** | -0.020** |
| boardsize | -0.017* | -0.026*** | -0.024*** | 0.266*** | 0.115*** | 0.016* | 0.027*** |
| ind | 0.0100 | 0.016* | 0.0110 | 0.053*** | 0.0100 | -0.0140 | -0.00700 |
| mngmhldn | 0.042*** | 0.036*** | 0.031*** | -0.141*** | -0.123*** | 0.086*** | 0.047*** |
| sharesbalance | 0.031*** | 0.017* | 0.024*** | 0.00900 | -0.031*** | -0.0130 | -0.0150 |
| dual | 0.0130 | 0.0100 | 0.015* | -0.085*** | -0.042*** | 0.00800 | 0.00100 |

| | pro | boardsize | ind | mngmhldn | sharesbalance | dual |
|---|---|---|---|---|---|---|
| pro | 1 | | | | | |
| boardsize | 0.209*** | 1 | | | | |
| ind | -0.021** | -0.415*** | 1 | | | |
| mngmhldn | -0.379*** | -0.142*** | 0.032*** | 1 | | |
| sharesbalance | -0.228*** | 0.071*** | -0.046*** | 0.203*** | 1 | |
| dual | -0.239*** | -0.144*** | 0.089*** | 0.196*** | 0.086*** | 1 |

Note: ***, **, * represent the significance level of 1%, 5%, 10%, respectively.

**Table 4. Baseline regression.**

| VARIABLES | (1) NCSKEW | (2) NCSKEW | (3) DUVOL | (4) DUVOL |
|---|---|---|---|---|
| sim_1 | 0.174*** | 0.102** | 0.118*** | 0.058** |
| | (4.28) | (2.40) | (4.40) | (2.07) |
| size | -0.017*** | -0.010 | -0.019*** | -0.016*** |
| | (-2.92) | (-1.53) | (-4.92) | (-3.70) |
| lev | -0.019 | 0.002 | -0.008 | 0.015 |
| | (-0.47) | (0.03) | (-0.30) | (0.49) |
| roa | 0.478*** | 0.388*** | 0.187** | 0.134 |
| | (3.43) | (2.71) | (2.04) | (1.43) |
| pro | -0.039** | -0.029* | -0.019* | -0.014 |
| | (-2.56) | (-1.80) | (-1.90) | (-1.29) |
| boardsize | 0.002 | -0.000 | 0.000 | -0.000 |
| | (0.48) | (-0.04) | (0.17) | (-0.13) |
| ind | 0.002 | 0.001 | 0.002* | 0.001 |
| | (1.40) | (1.07) | (1.94) | (1.60) |
| mngmhldn | 0.001* | 0.001* | 0.001 | 0.001 |
| | (1.88) | (1.74) | (1.59) | (1.36) |
| sharesbalance | 0.029** | 0.024* | 0.009 | 0.005 |
| | (2.30) | (1.88) | (1.11) | (0.61) |
| dual | -0.005 | -0.003 | -0.007 | -0.006 |
| | (-0.29) | (-0.15) | (-0.61) | (-0.54) |
| Constant | -0.144 | -0.002 | 0.074 | 0.145 |
| | (-1.13) | (-0.02) | (0.88) | (1.45) |
| Year FE | NO | YES | NO | YES |
| Industry FE | NO | YES | NO | YES |
| Observations | 12,168 | 12,168 | 12,168 | 12,168 |
| R-squared | 0.007 | 0.041 | 0.007 | 0.043 |
| F test | 0 | 0 | 0 | 0 |
| r2_a | 0.00607 | 0.0338 | 0.00611 | 0.0355 |
| F | 8.432 | 5.434 | 8.477 | 5.665 |
| Constant | -0.144 | -0.002 | 0.074 | 0.145 |

Note: ***, **, * represent the significance level of 1%, 5%, 10%, respectively.

average of 0.102 and 0.058 units, which has important practical significance for preventing the risk of stock price collapse.

**5.3.2 The impact of audit institution quality on inert information disclosure and the risk of stock price collapse.** Table 5 shows the group regression results of the quality moderating effect of audit institutions. From Table 5, it can be seen that in the regression analysis results of companies audited by non major international accounting firms (Big4 = 0), the coefficient of similarity in annual report texts is significantly positive at the 5% level. In the regression results of companies audited by the four major international accounting firms (Big4 = 1), the coefficient of similarity in annual report texts is positive but not significant. These results indicate that the positive correlation between the similarity of annual report text and the risk of stock price collapse significantly decreases under high-quality audit institutions, that is, key audit institutions weaken the impact of inert information disclosure on the risk of stock price collapse, verifying hypothesis H2.

**Table 5. Moderating effect.**

| VARIABLES | (1) NCSKEW | (2) NCSKEW | (3) DUVOL | (4) DUVOL |
|---|---|---|---|---|
| | Big4 = 0 | Big4 = 1 | Big4 = 0 | Big4 = 1 |
| sim_1 | 0.108** | 0.012 | 0.063** | 0.000 |
| | (2.38) | (0.10) | (2.11) | (0.00) |
| Constant | -0.034 | -0.348 | 0.107 | 0.059 |
| | (-0.20) | (-0.59) | (0.97) | (0.15) |
| controls | YES | YES | YES | YES |
| Year FE | YES | YES | YES | YES |
| Industry FE | YES | YES | YES | YES |
| Observations | 11,168 | 1,000 | 11,168 | 1,000 |
| R-squared | 0.041 | 0.143 | 0.043 | 0.146 |
| F test | 0 | 6.79e-07 | 0 | 2.68e-07 |
| r2_a | 0.0326 | 0.0745 | 0.0347 | 0.0779 |
| F | 4.886 | 2.087 | 5.139 | 2.140 |
| Constant | -0.034 | -0.348 | 0.107 | 0.059 |
| | (-0.20) | (-0.59) | (0.97) | (0.15) |

Note: ***, **, * represent the significance level of 1%, 5%, 10%, respectively.

## 5.4 Robustness test

**5.4.1 Measurement method for replacing independent variables.** The previous text adopts the similarity (sim_1) of the annual report text compared to the previous period to measure inert information disclosure and verify its positive correlation with the risk of stock price collapse. Therefore, when making inert information disclosure, will management not only imitate the previous annual report text information, but also refer to previous years' annual report information? Considering the implementation of the new "Enterprise Accounting Standards" in 2007, this article uses the 2007 annual report text information as the reference period and adopts the similarity of the annual report text compared to the 2007 annual report (sim_2007) to measure inert information disclosure. The regression results are shown in Table 6. There is a positive correlation between the similarity of the annual report text compared to the benchmark 2007 annual report and the risk of stock price collapse at the 5% level. Specifically, under the control of year and industry, when the similarity of the annual report text compared to the 2007 annual report increases by 1 unit, the average risk of stock price collapse increases by 0.163 and 0.093 units. At the same time, it can be seen that when the similarity of the annual report text compared to the previous period increases by one unit, the average risk of stock price collapse increases by 0.102 and 0.058 units, indicating that when the management tends to imitate the information in the benchmark annual report, the degree of increase in stock price collapse risk is greater.

**5.4.2 Change the sample interval.** In 2015, the Chinese stock market experienced a large-scale stock price crash, seriously affecting the normal operation of the capital market. To avoid the impact of stock price collapse on the relationship between inert information disclosure and stock price collapse risk, this article deleted the 2015 sample and re regressed according to model (4). The regression results are shown in Table 7. After deleting the sample of 2015, the similarity of the annual report text is still significantly positively correlated with the risk of stock price crash, which proves the robustness of the research conclusions in this paper.

**Table 6. Change the measurement method of independent variables.**

| VARIABLES | (1) ncskew | (2) ncskew | (3) duvol | (4) duvol |
|---|---|---|---|---|
| sim_1 | 0.102** | | 0.058** | |
| | (2.40) | | (2.07) | |
| sim_2007 | | 0.163** | | 0.093** |
| | | (2.40) | | (2.09) |
| Constant | -0.002 | -0.019 | 0.145 | 0.135 |
| | (-0.02) | (-0.12) | (1.45) | (1.34) |
| Controls | YES | YES | YES | YES |
| Year Fe | YES | YES | YES | YES |
| Industry FE | YES | YES | YES | YES |
| Observations | 12,168 | 12,168 | 12,168 | 12,168 |
| R-squared | 0.041 | 0.041 | 0.043 | 0.043 |
| F test | 0 | 0 | 0 | 0 |
| r2_a | 0.0338 | 0.0338 | 0.0355 | 0.0355 |
| F | 5.434 | 5.434 | 5.665 | 5.666 |

Note: ***, **, * represent the significance level of 1%, 5%, 10%, respectively.

## 6 Further analysis

It is of great significance to deeply analyze the root of lazy information disclosure on the basis of the clear positive impact of lazy information disclosure on stock price crash risk. On the one hand, based on the separation of management rights and ownership in the modern enterprise system, the inconsistency between the interests of shareholders and management will lead to adverse selection and moral hazard of management. Adverse selection refers to the existence of information asymmetry between shareholders and management. The management makes use of the existing information advantages to conduct insider trading and obtain excess profits to damage the interests of shareholders. In order to avoid being discovered, the management tends to use the previously recognized annual report information. Moral hazard is when

**Table 7. Change the sample interval.**

| VARIABLES | (1) ncskew_mdtl | (2) ncskew | (3) duvol | (4) duvol |
|---|---|---|---|---|
| sim_1 | 0.154*** | 0.093** | 0.113*** | 0.057** |
| | (3.60) | (2.09) | (4.04) | (1.96) |
| Constant | -0.295** | -0.152 | -0.021 | 0.055 |
| | (-2.21) | (-0.96) | (-0.24) | (0.53) |
| Controls | YES | YES | YES | YES |
| Year Fe | YES | YES | YES | YES |
| Industry FE | YES | YES | YES | YES |
| Observations | 11,252 | 11,252 | 11,252 | 11,252 |
| R-squared | 0.008 | 0.042 | 0.007 | 0.044 |
| F test | 0 | 0 | 0 | 0 |
| r2_a | 0.00680 | 0.0334 | 0.00654 | 0.0361 |
| F | 8.706 | 5.090 | 8.409 | 5.440 |

Note: ***, **, * represent the significance level of 1%, 5%, 10%, respectively.

management does not work hard and does not aim to maximize the interests of shareholders, but aims to maximize their leisure time and compensation. Both adverse selection and moral hazard assume that management has an incentive to adopt lazy forms of information disclosure. Based on the above analysis, lazy information disclosure may come from the management's "intentional action". On the other hand, listed companies disclose the text of annual report to reflect the company's financial status, operating results, future development prospects and other information. If the company's current development situation is no significant progress compared to the past, management tends to stick to the text of the previous annual report. Based on the above analysis, lazy information disclosure may also come from business situation "the fact is so".

In order to verify whether the lazy information disclosure comes from the management's "intentional action" or the business situation "as it is", this paper first assumes that the lazy information disclosure comes from the business situation "the fact is so". In view of the fact that high analyst attention and research report tracking are conducive to improving the overall information quality of listed companies, it is less likely for management to use the annual report information of previous years. In this paper, regression is performed after grouping according to the median anaattention and reportattention of analysts. Table 8 Regression results show that: in the case of companies with a large number of analysts' attention and more research reports tracking, that is, in the absence of self-interested motives of the management, the similarity of the annual report text is significantly positively correlated with the risk of stock price crash, which excludes the possibility that the inert information disclosure comes from the management's "intentional action"", indicating that the inert information disclosure comes from the business situation "the fact is so".

The annual report of the listed company can be published only after it is compiled by the management, audited by the accounting firm and reviewed by the board of directors. Reporting time lag refers to the interval between the date of assets and liabilities and the time of annual report disclosure. Companies with short information disclosure delay have less time for management to implement opportunistic behaviors, which precludes the possibility of management hiding bad news from the side. In order to improve the quality of information disclosure of listed companies and enhance the transparency of the securities market, the

**Table 8. Regression results of controlling the quality of information disclosure.**

| VARIABLES | (1) | (2) | (3) | (4) |
|---|---|---|---|---|
| | (anaattention) | | (reportattention) | |
| | Less attention | More attention | Less attention | More attention |
| sim_1 | 0.095 | 0.105* | 0.099 | 0.102* |
| | (1.51) | (1.88) | (1.51) | (1.88) |
| Constant | 0.667*** | -0.235 | 0.746*** | -0.352 |
| | (2.66) | (-1.01) | (2.85) | (-1.56) |
| Controls | YES | YES | YES | YES |
| Year FE | YES | YES | YES | YES |
| Observations | 6,521 | 5,647 | 6,118 | 6,050 |
| R-squared | 0.050 | 0.069 | 0.052 | 0.064 |
| F test | 0 | 0 | 0 | 0 |
| r2_a | 0.0357 | 0.0532 | 0.0373 | 0.0488 |
| F | 3.542 | 4.372 | 3.518 | 4.235 |

Note: ***, **, * represent the significance level of 1%, 5%, 10%, respectively.

**Table 9. Regression results of controlling information disclosure time.**

| VARIABLES | (1) | (2) | (3) | (4) | (5) | (6) |
|---|---|---|---|---|---|---|
| | (totallaga) | | (totallagb) | | (put_off) | |
| | Short time | Long time | Short time | Long time | NO | YES |
| sim_1 | 0.178*** | 0.030 | 0.161*** | 0.051 | 0.113** | 0.011 |
| | (3.09) | (0.47) | (2.78) | (0.82) | (2.47) | (0.10) |
| Constant | 0.038 | -0.082 | 0.084 | -0.107 | -0.077 | 0.021 |
| | (0.17) | (-0.36) | (0.38) | (-0.47) | (-0.46) | (0.05) |
| Controls | YES | YES | YES | YES | YES | YES |
| Year FE | YES | YES | YES | YES | YES | YES |
| Observations | 6,277 | 5,891 | 6,126 | 6,042 | 10,442 | 1,726 |
| R-squared | 0.060 | 0.042 | 0.061 | 0.041 | 0.043 | 0.102 |
| F test | 0 | 0 | 0 | 0 | 0 | 1.25e-07 |
| r2_a | 0.0450 | 0.0259 | 0.0459 | 0.0253 | 0.0339 | 0.0510 |
| F | 4.083 | 2.649 | 4.068 | 2.649 | 4.776 | 2.008 |

Note: ***, **, * represent the significance level of 1%, 5%, 10%, respectively.

annual report appointment system of listed companies has been implemented since 2002. If the company discloses the annual report after the appointment date, there is the possibility of management manipulation. To this end, this paper is divided into groups according to the median of annual report delay (totallaga) in calculating holidays and annual report delay (totallagb) in not calculating holidays and put_off or not, so as to further verify the source of lazy information disclosure. The regression results are shown in Table 9. The positive correlation between the similarity of the annual report text and the risk of stock price crash shows a significant positive correlation between the information disclosure delay is short and the group disclosed within the appointment time, that is, the similarity of the annual report text and the risk of stock price crash are more significant when the management lacks the time to take self-interested behaviors. It excludes the possibility that the lazy information disclosure comes from the management's "intentional action", and further indicates that the lazy information disclosure comes from the business situation "the fact is so".

## 7 Conclusion and enlightenment

This paper takes China's A-share listed companies from 2011 to 2022 as the research sample, takes the annual report text disclosure form as the entry point, uses the computer text analysis technology to measure the lazy information disclosure, and explores the relationship between it and the stock price crash risk. The results show that: first of all, the similarity of annual report text has a significant positive correlation with the risk of stock price crash, that is, when the information of annual report text is presented in the form of lazy information disclosure, the risk of stock price crash increases. Second, for companies audited by key auditing institutions, the positive correlation between the similarity of their annual reports and the risk of stock price crash is not significant, indicating that key auditing institutions will weaken the positive correlation between lazy information disclosure and the risk of stock price crash. Further, through the analysis of the external attention and the time delay of annual report disclosure, it is concluded that the management lacks the opportunity and time to hide the bad news, so it is clear that the lazy information disclosure comes from the business situation "the fact is so". After replacing the measurement method of lazy information disclosure and

changing the sample interval, lazy information disclosure is still significantly positively correlated with stock price crash risk, indicating that the research conclusion has a certain reliability.

This study takes the form of information disclosure in the text of the annual report as the starting point to study the stock price crash risk, to build a bridge between the form of information disclosure and the stock price crash risk, and to clarify the relationship between lazy information disclosure and the stock price crash risk. And with the help of the computer text analysis technology to calculate the annual report text similarity to measure the degree of lazy information disclosure on the basis of quantitative lazy information disclosure to provide a new perspective for the annual report text analysis. In addition, in-depth analysis of the source of lazy information disclosure, deepening the form of information disclosure related research. The research conclusions of this paper are of great practical significance for improving the information disclosure policy, preventing and controlling the risk of stock price crash, and facilitating the mature development of China's capital market in the new era. At the same time, it also provides useful references for the improvement and development of emerging capital markets. From the perspective of listed companies, in view of the empirical test finding that lazy information disclosure can reduce information transparency and thus increase the risk of stock price crash, listed companies should pay more attention to "usefulness" in the disclosure process of annual reports, highlighting the information important to investors' decision-making, and cutting down the redundant information that investors do not care about or have disclosed in previous years, so as to improve information transparency. At the same time, listed companies should pay real-time attention to the similarity of the text of the annual report, timely assess the risk of stock price crash of listed companies, and quickly take evasive measures when the risk of stock price crash rises. In addition, listed companies should hire key audit institutions to provide a strong authentication role for the company's annual report and alleviate the information asymmetry between investors, analysts and other information users and the company. From the perspective of government departments, the government and other regulatory departments should formulate high-standard information disclosure policies, strictly supervise the information disclosure behavior of listed companies, reduce the phenomenon of irregular information disclosure and frequent information correction, effectively improve the information asymmetry, and improve the quality of information disclosure. And carry out quality evaluation of information disclosure of listed companies, urge them to pay attention to information disclosure work, encourage voluntary information disclosure of listed companies, and improve the information listing of public shareholders. Based on the perspective of stakeholders, investors, analysts and other stakeholders should broaden the channels of information acquisition, and make full use of the Internet to collect other information related to the enterprise on the basis of paying attention to the annual report, such as reports issued by research institutions, news reports, etc., to have an in-depth understanding of the company's financial status and operating results, correctly evaluate the value of the company, and realize efficient decision-making.

This article adopts a text vectorization method to calculate the similarity of annual report texts and uses it to measure inert information disclosure. It is worth noting that the appearance of the same features in different positions and order indicates different meanings. Text vectorization only considers the number of times certain features appear in the text, without considering their position and order. In addition, text vectorization cannot recognize similarities at the semantic level, and cannot effectively recognize texts with different fonts but similar meanings. In future research, unsupervised machine learning methods can be adopted to reduce the imprecision of text vectorization methods due to the drawbacks of polysemy, semantic ambiguity, and not considering the order and position of words.

## Supporting information

**S1 File. Sample data.**
(XLSX)

## Author Contributions

**Conceptualization:** Xiaofei Shi, Xuefen Cao.

**Data curation:** Wenxin Xu.

**Formal analysis:** Yangshi Hou.

**Funding acquisition:** Xiaofei Shi.

**Investigation:** Liwei Shang.

**Methodology:** Xiaofei Shi.

**Project administration:** Liwei Shang.

**Resources:** Xiaofei Shi.

**Software:** Xuefen Cao.

**Supervision:** Xiaofei Shi.

**Validation:** Xiaofei Shi, Xuefen Cao, Wenxin Xu.

**Visualization:** Wenxin Xu.

**Writing – original draft:** Xuefen Cao.

**Writing – review & editing:** Xiaofei Shi.

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
