## [Decision Letter · Decision Letter 0]

22 Jun 2023

PONE-D-23-15931The Influence of lazy information disclosure on stock price crash risk: Empirical evidence from ChinaPLOS ONE

Dear Dr. Shi,

Thank you for submitting your manuscript to PLOS ONE. After careful consideration, we feel that it has merit but does not fully meet PLOS ONE’s publication criteria as it currently stands. Therefore, we invite you to submit a revised version of the manuscript that addresses the points raised during the review process.

We look forward to receiving your revised manuscript.

Kind regards,

Rana Muhammad Ammar Zahid, PhD

Academic Editor

PLOS ONE

Journal Requirements:

"This research was funded by National Natural Science Foundation of China [72172063]; S&T Program of Hebei [22557607D]; Humanities and Social Science Research Project of Hebei Education Department [SD2022054]; The Project of Corporate Governance and Enterprise Research Center [GS2023G]."

4. Please clarify the Table 2 "Descriptive statistics." in page "9" and Table 2 "correlation analysis" in page "10".

Additional Editor Comments:

Dear Authors,

After careful consideration and review, we request minor amendments to this manuscript before accepting it for publication.

Please answer the reviewers comments to improve the manuscript, as it has area to improve in all section, introduction, hypothesis development, methodology and conclusion. Moreover, there is an important strand of literature with respect to the financial reporting quality and stock markets as discussed in following studies. These studies will help you to improve the methodology as well.

The impact of International Financial Reporting Standards (IFRS) adoption on the integration of capital markets

International Journal of Finance & Economics, 1– 22.

https://doi.org/10.1002/ijfe.2684

An Analysis of IFRS and SME-IFRS Adoption Determinants: A Worldwide Study

Emerging Markets Finance and Trade, 55:2, 391-408.

https://doi.org/10.1080/1540496X.2018.1500890

Make sure to proofread the manuscript before it is resubmitted to the journal. Please go through the journal’s guidelines thoroughly and revise the paper accordingly. Thank you for submitting your paper to the PlosOne.

Reviewers' comments:

Reviewer's Responses to Questions

**Comments to the Author**

1. Is the manuscript technically sound, and do the data support the conclusions?

Reviewer #1: Partly

Reviewer #2: Yes

2. Has the statistical analysis been performed appropriately and rigorously? 

Reviewer #1: Yes

Reviewer #2: Yes

3. Have the authors made all data underlying the findings in their manuscript fully available?

Reviewer #1: Yes

Reviewer #2: Yes

4. Is the manuscript presented in an intelligible fashion and written in standard English?

Reviewer #1: Yes

Reviewer #2: Yes

5. Review Comments to the Author

Reviewer #1: 1) In the section of Research on influencing factors of stock price crash risk, the authors need to refer to macroeconomic elements as one of influencing factors. For example, the below research can be an ideal example of this topic. https://doi.org/10.3390/su13073688

2)In 2023, various researches have been conducted on the risk of falling stock prices in different markets, why haven't the authors addressed them?

3) The research methodology is relatively well designed in such a way that most of the important issues have been paid attention to.

4)In the conclusion section, why the authors have not clearly addressed the limitations of their research in the Chinese market. Authors are also better off presenting their research application proposals to other researchers.

Reviewer #2: This paper discusses the economic consequences of lazy annual report information disclosure, which has certain novelty.

The following aspects are for the author to further improve the paper reference:

#Q1 The article does not clearly explain why lazy information disclosure exacerbated the stock price crash. In 3.1 Lazy information disclosure and stock price crash risk, the author explains that lazy information reduces information content and increases stock price synchronicity, which is acceptable. However, this paper does not test the influence of lazy information disclosure on stock price synchronicity, nor does it use stock price synchronicity as an intermediate variable to test stock price crash risk.

In addition, the author emphasizes in the article: When the stock index is in the rising cycle, the synchronicity of stock price makes the stock price overvalued, and the stock price is inconsistent with the actual value of the company. When the stock index is in a downward cycle, the synchronicity of stock price will lower the stock price, and the management tends to release false news to maintain the stock price, so that the stock price deviates from the actual value of the company and increases the possibility of stock price crash.

The contents emphasized above are not controlled and demonstrated in the empirical part, which is too subjective and far-fetched.

Therefore, it is suggested that the author consider the direct impact of Lazy information disclosure on stock price crash risk from the classical hypothesis of stock price crash risk itself. For example, can we consider the hypothesis of managing bad news hiding put forward by Jin and Myesr(2006) and clearly demonstrate the possible explanation of Lazy information disclosure for bad news hiding?

#Q2 In part 3.2 The regulating role of audit institutions, the paper also puts forward "the quality of the annual report information". The following three details are not clear. This hypothesis is also not strongly supported.

#Q2.1 The article also did not clearly account for the relationship between Lazy information disclosure and the quality of the annual report information.

#Q2.2 If big auditors can improve "the quality of the annual report information", then why do companies audited by big auditors still have "Lazy information disclosure"?

#Q2.3 How the "Lazy information disclosure" after audit by a large auditors differs from other samples.

#Q3 4.2.2 Explain variables Whether there are relevant literatures for reference.

#Q4 This paper tests whether it is appropriate for the relevant variables of financial report to lag 2 periods in the model, and it is common to lag 1 period.

6. PLOS authors have the option to publish the peer review history of their article (what does this mean?). If published, this will include your full peer review and any attached files.

Reviewer #1: **Yes: **Hossein Tarighi

Reviewer #2: No

---

## [Author Response · Author response to Decision Letter 0]

28 Jun 2023

Thank you for the reviewers’ comments concerning our manuscript entitled “The Influence of lazy information disclosure on stock price crash risk: Empirical evidence from China” [PONE-D-23-15931]. Those comments are all valuable and very helpful for revising and improving our paper, as well as the important guiding significance to our researches. We have studied comments carefully and have made corrections which we hope meet with approval. Revised portion are marked in red in the paper. The main corrections in the paper and the responds to the reviewer’s comments are as following.

Response to academic editor

(1)Revise the paper according to PLOS ONE's style requirements.

(2)The funders had no role in study design, data collection and analysis, decision to publish, or preparation of the manuscript. 

(3)The data used in this article have upload as Supporting Information files.

(4)Table 2 on page 9 is a descriptive statistical analysis, and Table 3 on page 10 is a correlation analysis, which have been corrected. 

(5)We have checked and ensured that the cited references are correct and complete. 

(6)We have read and cited important references related to this study that you have proposed. 

Response to Reviewer 1 Comments

1)In the section of Research on influencing factors of stock price crash risk, the authors need to refer to macroeconomic elements as one of influencing factors. For example, the below research can be an ideal example of this topic. https://doi.org/10.3390/su13073688

Thank you for your comments on our research. Your comments are of great significance to our paper writing and scientific research. Considering your suggestion, we have added relevant literature on the impact of macro factors on the risk of stock price collapse in the literature review section. The following is the added content in the literature review section. (The relevant contents are listed in lines 116 to121 in the text)

External factors such as inflation, unemployment rate, GDP, and exchange rate (Mahdi et al., 2021) , institutional environment (Li, 2018), and macro factors such as capital market opening up (Li and Xu, 2019) ; Macro policies such as industrial policy tools (Wu et al., 2023) , merger of national and local taxes (Chen et al., 2023) , and green credit (Zhang and Jiang, 2022).

2)In 2023, various researches have been conducted on the risk of falling stock prices in different markets, why haven't the authors addressed them?

Thank you for your comments on our research. Your comments are of great significance to our paper writing and scientific research. Considering your suggestion, We have added the latest research findings on the risk of stock price collapse in 2023 in the literature review section. The following is the added content in the literature review section. (The relevant contents are listed in lines 112 to 121 in the text)

Existing literature divides the influencing factors of stock price crash risk into internal factors and external factors. Internal factors such as the behavior of managers (3, 4), Characteristics of managers (5), quality of accounting information (6), corporate governance (7, 8), mixed ownership reform (9), digital transformation (10) green transformation of enterprises (11), ESG performance (12, 13), etc. External factors such as inflation, unemployment rate, GDP, and exchange rate (14), institutional environment (15), and capital market openness (16), among other macro factors; Macro policies such as industrial policy tools (17), merger of national and local taxes (18), and green credit (19); Stakeholders such as securities analysts (20) and institutional investors (21, 22). All the above factors can be explained by the information asymmetry theory (23) and the "bad news" hiding hypothesis (24).

3)The research methodology is relatively well designed in such a way that most of the important issues have been paid attention to.

Thanks for the reviewer's recognition of the research methodology. In future research, we will learn more scientific methods to improve the accuracy and robustness of the research.

4)In the conclusion section, why the authors have not clearly addressed the limitations of their research in the Chinese market. Authors are also better off presenting their research application proposals to other researchers.

Thank you for your comments on our research. Your comments are of great significance to our paper writing and scientific research. Considering your suggestion, we added research limitations and prospects at the end of the article. (The relevant contents are listed in lines 517 to 525 in the text)

This article adopts a text vectorization method to calculate the similarity of annual report texts and uses it to measure inert information disclosure. It is worth noting that the appearance of the same features in different positions and order indicates different meanings. Text vectorization only considers the number of times some features appear in the text, and does not consider the position and order of their appearance. In addition, text vectorization cannot recognize similarities at the semantic level, and cannot effectively recognize texts with different fonts but similar meanings. In future research, unsupervised machine learning methods can be adopted to reduce the imprecision of text vectorization methods due to the drawbacks of polysemy, semantic ambiguity, and not considering the order and position of words.

Response to Reviewer 2 Comments

#Q1 The article does not clearly explain why lazy information disclosure exacerbated the stock price crash. In 3.1 Lazy information disclosure and stock price crash risk, the author explains that lazy information reduces information content and increases stock price synchronicity, which is acceptable. However, this paper does not test the influence of lazy information disclosure on stock price synchronicity, nor does it use stock price synchronicity as an intermediate variable to test stock price crash risk.

In addition, the author emphasizes in the article: When the stock index is in the rising cycle, the synchronicity of stock price makes the stock price overvalued, and the stock price is inconsistent with the actual value of the company. When the stock index is in a downward cycle, the synchronicity of stock price will lower the stock price, and the management tends to release false news to maintain the stock price, so that the stock price deviates from the actual value of the company and increases the possibility of stock price crash.

The contents emphasized above are not controlled and demonstrated in the empirical part, which is too subjective and far-fetched.

Therefore, it is suggested that the author consider the direct impact of Lazy information disclosure on stock price crash risk from the classical hypothesis of stock price crash risk itself. For example, can we consider the hypothesis of managing bad news hiding put forward by Jin and Myesr(2006) and clearly demonstrate the possible explanation of Lazy information disclosure for bad news hiding?

Thank you for your comments on our research. Your comments are of great significance to our paper writing and scientific research. Considering your suggestion, this article conducts a theoretical analysis based on the "bad news concealment" hypothesis: lazy information disclosure reduces information content, reduces annual report information transparency, and provides opportunities for management to conceal bad news. When bad news accumulates to a certain extent and erupts, investors will sell stocks in large quantities, increasing the risk of stock price collapse. The specific theoretical analysis is as follows:

The annual reports of listed companies are the main source of information for a large number of investors to make decisions, and accurate and complete information plays a crucial role in the decision-making process of investors. Given that the annual report carries information about the company's operating status and future development direction, detailed textual explanations help investors accurately assess the company's value, thereby reducing the risk of decision-making errors caused by information asymmetry (45, 46). Therefore, the information conveyed by the annual report text plays an important role in the stability of stock prices.

Annual report as a specific performance of information disclosure by the company's management, the form of its text information disclosure may have an impact on the characteristics of the text information of the annual report. Because the lazy information disclosure has the characteristics of template and does not change with time inertia, when the text information of the current report is expressed in the form of lazy information disclosure, the text of the company's annual report has higher similarity, that is, the current annual report updates less than the previous year's annual report, and the text is closer to each other. Inert information disclosure indicates that the management largely uses previous annual report information during information disclosure and does not truthfully convey information about the current operating status and future development direction. This means that the current annual report has less updated content compared to previous annual reports, which to some extent reduces the transparency of annual report information. The decline in information transparency provides an opportunity for management to conceal bad news (47). Although the management needs to regularly disclose the company's financial condition, operating results, and other company specific information in accordance with the "Management Measures for Information Disclosure of Listed Companies". However, it is worth noting that existing research suggests that management, driven by self-interest motivation, may conceal bad news in order to obtain equity incentives (48) and excess returns (3). At the same time, as the direct responsible person for the annual report text, the management has a certain right to choose the content and timing of information disclosure, especially when the listed company is not doing well or the investment project risk is high, the management may use their private rights to conceal bad news in order to protect their own interests.

When management adopts lazy information disclosure, the decrease in information transparency and self-interest motivation of management increase the possibility of management hiding bad news. Investors are prone to make unreasonable judgments about the company's value without being familiar with its financial situation, operating results, and development prospects. However, the concealment of bad news requires a certain cost, and when the management cannot bear the cost of concealing bad news, negative news tends to erupt. At this point, investors may sell a large amount of stocks, leading to a sharp drop in stock prices (49). 

#Q2 In part 3.2 The regulating role of audit institutions, the paper also puts forward "the quality of the annual report information". The following three details are not clear. This hypothesis is also not strongly supported.

#Q2.1 The article also did not clearly account for the relationship between Lazy information disclosure and the quality of the annual report information.

Thank you for your comments on our research. Your comments are of great significance to our paper writing and scientific research. The following is our explanation of the problem.

When the management adopts an inert information disclosure strategy, the similarity of the annual report text is higher, that is, the effective information content carried by the annual report text is lower. The integrity of the annual report text information has been weakened, which to some extent increases the information asymmetry between external stakeholders and the company, which is not conducive to investors and analysts' judgment of the company's value. Therefore, this article believes that lazy information disclosure will reduce the quality of annual report text information.

#Q2.2 If big auditors can improve "the quality of the annual report information", then why do companies audited by big auditors still have "Lazy information disclosure"?

Thank you for your comments on our research. Your comments are of great significance to our paper writing and scientific research. The following is our explanation of the problem.

External audits constrain the behavior of management to a certain extent by auditing financial reports, while providing independent assurance to investors to ensure that the company's financial statements comply with generally accepted accounting principles, thereby ensuring the quality of annual report information and reducing the degree of information asymmetry between management and external shareholders . Key audit institutions have a large scale, high level of auditor expertise, strong independence, and the ability to objectively and fairly evaluate the company's annual reports. Compared to non key audit institutions, key audit institutions have a stronger role in ensuring the quality of annual report information. However, external audit only serves as a third-party institution and has the responsibility of reviewing the company's annual report. Inert information disclosure is a disclosure method adopted by management, and financial statement auditing cannot alleviate the responsibilities of the management and governance of the audited entity. Therefore, companies audited by large accounting firms still have inert information disclosure.

#Q2.3 How the "Lazy information disclosure" after audit by a large auditors differs from other samples.

Thank you for your comments on our research. Your comments are of great significance to our paper writing and scientific research. We will explain this issue from both theoretical and empirical perspectives.

Theoretical analysis: The scale of key audit institutions is relatively large, and auditors have a high level of professional competence and strong independence. They are able to objectively and fairly evaluate the company's annual reports. On the one hand, the audit opinions issued by key audit institutions have a stronger verification effect, which largely guarantees the reliability of annual report information disclosure (53). On the other hand, the selection of key audit institutions means that the company is willing to promise to provide reliable annual report information. When the current report is presented in the form of lazy information disclosure, Key audit institutions audit can alleviate the information asymmetry between stakeholders and companies by sending positive signals to the outside world. On the one hand, the annual report as a direct channel for investors to understand the listed company, the annual report authenticated by key audit institutions helps investors to make accurate judgments, thus reducing the risk of stock price collapse. On the other hand, the annual report is an important source of analysts' forecasts. The annual report authenticated by key audit institutions helps attract more analysts' attention, improving the accuracy of analysts' forecasts, and then reducing the synchronicity of stock prices and reduce the risk of stock price crash (54).

Empirical testing: To explore the differences between the samples audited by key audit institutions and other samples, this article conducts a T-test based on whether the audit institution belongs to a key audit institution. The T-test results are shown in Table 1. Through the T-test results, it can be seen that there are significant differences in the risk of stock price collapse among the sample companies audited by different audit institutions. And the risk of stock price collapse in the samples audited by key audit institutions is significantly lower than that in the samples audited by non key audit institutions.

Table 1. two sample t-test

Group Obs Mean Std. err. Std. dev. [95%conf. interval]

0 11,168 -.2201561 .0044664 .4720077 [-.2289112 -.2114011]

1 1,000 -.2511788 .0150073 .4745721 [-.2806282 -.2217294]

Combined 12,168 -.2227057 .0042814 .4722762 [-.2310979 -.2143134]

 .0310226 .0155871 [.0004695 .0615758]

diff = mean(0)-mean(1) t = 1.9903

H0: diff = 0 Degrees of freedom = 12166

Ha: diff < 0 Ha: diff != 0 Ha: diff > 0

Pr(T < t) = 0.9767 Pr(|T|> t) = 0.0466 Pr(T > t) = 0.0233

#Q3 4.2.2 Explain variables Whether there are relevant literatures for reference.

Thank you for your comments on our research. Your comments are of great significance to our paper writing and scientific research. The following is our explanation of the problem.

Considering that when the annual report text is presented in the form of inert information disclosure, the similarity of the annual report text is relatively high, so this article uses the similarity of the annual report text to measure the degree of inert information disclosure.This article refers to the research of Zheng and Liu (2022) and adopts a text vectorization method to calculate the similarity of annual report texts. The calculation process is as follows: (1) Common stop words such as Baidu Stop word, Harbin Institute of Technology Stop word, and Sichuan University Machine Intelligence Laboratory words are used to stutter the text; (2) Clean the segmentation results; (3) Calculate the TF-IDF value of words in the text. TF refers to the frequency of words appearing in the text, and IDF refers to the probability of a certain word appearing in all texts. The adjusted word frequency is the product of TF and IDF, giving higher weights to words that better reflect text features; (4) Calculate text similarity using cosine function.

#Q4 This paper tests whether it is appropriate for the relevant variables of financial report to lag 2 periods in the model, and it is common to lag 1 period.

Thank you for your comments on our research. Your comments are of great significance to our paper writing and scientific research. The following is our explanation of the problem.

Considering that the current annual report is often disclosed before May of the next issue, it takes some time for the information conveyed by the annual report text to be absorbed by the capital market, so research related to the information in the annual report text often lags behind one period of processing (Qian and Zhu, 2020). However, it is worth noting that this study examines the impact of information conveyed in annual report texts on the risk of stock price collapse. The fundamental reason for the risk of stock price collapse is the outbreak of negative news, and the accumulation of negative news also takes some time to be reflected in the market (Luo and Du , 2014). Therefore, this study will lag the variables related to the text information of the annual report for two periods.

---

## [Editor Report · Decision Letter 1]

29 Jun 2023

The Influence of lazy information disclosure on stock price crash risk: Empirical evidence from China

PONE-D-23-15931R1

Dear Dr. Shi,

We’re pleased to inform you that your manuscript has been judged scientifically suitable for publication and will be formally accepted for publication once it meets all outstanding technical requirements.

Kind regards,

Rana Muhammad Ammar Zahid, PhD

Academic Editor

PLOS ONE

Additional Editor Comments (optional):

Thank you for incorporating the suggestions.
---

## [Editor Report · Acceptance letter]

11 Aug 2023

PONE-D-23-15931R1 

The influence of lazy information disclosure on stock price crash risk: Empirical evidence from China 

Dear Dr. Shi:

I'm pleased to inform you that your manuscript has been deemed suitable for publication in PLOS ONE. Congratulations! Your manuscript is now with our production department. 

Kind regards, 

on behalf of

Dr. Rana Muhammad Ammar Zahid 

Academic Editor

PLOS ONE